# Induction of Invasive Basal Phenotype in Triple-Negative Breast Cancers by Long Noncoding RNA BORG

**DOI:** 10.3390/cancers16183241

**Published:** 2024-09-23

**Authors:** Farshad Niazi, Kimberly A. Parker, Sara J. Mason, Salendra Singh, William P. Schiemann, Saba Valadkhan

**Affiliations:** 1Department of Molecular Biology and Microbiology, Case Western Reserve University, Cleveland, OH 44106, USA; farshad.niazi@gmail.com (F.N.); sjm203@case.edu (S.J.M.); 2Department of Biochemistry, Case Western Reserve University School of Medicine, Cleveland, OH 44106, USA; kap134@case.edu; 3Center for Immunotherapy and Precision Immuno-Oncology (CITI), Lerner Research Institute, Cleveland Clinic, Cleveland, OH 44195, USA; sxs1528@case.edu

**Keywords:** BORG, lncRNAs, TNBC, cancer stem cells

## Abstract

**Simple Summary:**

Breast cancer remains a major health issue in the United States and around the world. While progress has been made in curing most types of breast cancer, there are currently no effective cures for a subtype called ‘triple-negative’ breast cancer. We have identified a gene named *BORG* that, based on experiments in cultured human cells and mice, plays an important role in triple-negative breast cancer. Here, we study the function of this gene in a large group of human triple-negative breast cancers and show that when the level of expression of this gene is high, tumors become more invasive and develop ‘triple-negative’-like properties. Even in non-triple-negative tumors, high levels of expression of this gene are associated with more aggressive tumor behavior and poor response to therapy. Based on these validation studies in human tumors, we hope to leverage mechanistic vulnerabilities in BORG action to one day advance novel therapeutic strategies to alleviate hard-to-cure triple-negative breast tumors.

**Abstract:**

Background/Objectives: Long noncoding RNAs (lncRNAs) are known to play key roles in breast cancers; however, detailed mechanistic studies of lncRNA function have not been conducted in large cohorts of breast cancer tumors, nor has inter-donor and inter-subtype variability been taken into consideration for these analyses. Here we provide the first identification and annotation of the human BORG lncRNA gene. Methods/Results: Using multiple tumor cohorts of human breast cancers, we show that while BORG expression is strongly induced in breast tumors as compared to normal breast tissues, the extent of BORG induction varies widely between breast cancer subtypes and even between different tumors within the same subtype. Elevated levels of BORG in breast tumors are associated with the acquisition of core cancer aggression pathways, including those associated with basal tumor and pluripotency phenotypes and with epithelial–mesenchymal transition (EMT) programs. While a subset of BORG-associated pathways was present in high BORG-expressing tumors across all breast cancer subtypes, many were specific to tumors categorized as triple-negative breast cancers. Finally, we show that genes induced by heterologous expression of BORG in murine models of TNBC both in vitro and in vivo strongly overlap with those associated with high BORG expression levels in human TNBC tumors. Conclusion: Our findings implicate human BORG as a novel driver of the highly aggressive basal TNBC tumor phenotype.

## 1. Introduction

Despite significant improvements in the early detection and treatment of breast cancers, this disease persists as a global health concern that ranks among the leading causes of mortality in women [1,2]. Breast cancers are highly heterogeneous at both the cellular and molecular levels and are classified clinically by immunohistochemistry and/or in situ hybridization-based detection of estrogen receptor-α (ER-α), progesterone receptor (PR), and HER2/ErbB-2 expression [3,4,5]. Importantly, the presence of these receptors continues to serve as some of the most effective therapeutic targets in all of oncology, particularly for patients harboring breast tumors belonging to the luminal A (ER-α- and PR-positive), luminal B (ER-α and HER2-positive), and HER2 subtypes [6]. In stark contrast, patients harboring triple-negative breast cancers (TNBCs), which lack notable expression of ER-α, PR, and HER2, exhibit the highest disease grades and proliferation indices that culminate in (i) the worst progression-free and overall survival rates of all breast cancer subtypes, and (ii) rapid relapse and early death within 5 years of initial diagnosis and treatment [6,7,8]. Despite the similarities needed to classify breast cancers into subtypes, there remains significant variability among individual human breast tumors, even those from the same subtype. This, in turn, further complicates the choice of therapy and significantly affects patient response to treatment and overall prognosis. Transcriptomic profiling of TNBCs using multiple methods such as PAM50 [9] and TNBCtype-4 [10] finds that these tumors predominantly fall into the basal-like category, which are highly metastatic and robustly express genes typically found in basal/myoepithelial cells, such as cytokeratins 14, and 16 [11]. Moreover, TNBCs are also enriched in the expression of transcriptional programs coupled to breast cancer stem cells (e.g., CD44, ITGA6/α6 Integrin, ALDH1A1, and CD133/PROM1 [12,13,14,15,16]) and epithelial–mesenchymal transition (EMT) programs [6]. They are also characterized by (i) elevated expression of the receptors for epidermal growth factor (EGFR) and stem cell growth factor (KIT) [17,18,19]; and (ii) mutational inactivation of BRCA and p53, resulting in impaired DNA damage response and high mutational burden [20,21]. Finally, recurrent TNBCs frequently acquire resistance to standard-of-care chemotherapeutic agents (e.g., doxorubicin, cyclophosphamides, and taxanes) through mechanisms that remain incompletely understood. More recently, immunotherapy approaches (e.g., monoclonal antibodies against programmed death-ligand (PD-L1)) have made inroads into the breast oncology space, particularly for the treatment of TNBCs due to their higher mutational burden, higher infiltration of tumor lymphocytes, and higher expression of PD-L1 [7]. Unfortunately, this strategy has failed to significantly improve overall patient survival and outcomes in monotherapy settings [22]. This failure stems from our lack of understanding of the variability and complex nature of the molecular networks that drive the behavior of breast cancer tumors, even those from the same subtype, in a patient-specific manner. Clearly, a new paradigm in cancer research that takes tumor variability into consideration is urgently needed for the identification of new therapeutic strategies that can achieve curative outcomes for TNBC patients.

Long noncoding RNAs (lncRNAs) are a heterogeneous group of cellular RNAs that are transcribed by RNA polymerases I, II, or III, and thus, may or may not be spliced and polyadenylated [23,24,25]. Most lncRNAs are less evolutionarily conserved, expressed at lower copy numbers, and less efficiently spliced compared to protein-coding genes [26,27]. Nonetheless, these noncoding transcripts constitute the dominant output of the human genome in terms of the fraction of the genome comprising their loci [23,24,25]. Based on the latest definition of this class of molecules, lncRNAs range from around five hundred to tens of thousands of nucleotides in length, with many containing short open reading frames that may even code for short peptides. Nonetheless, they perform an RNA-mediated cellular function [23,24,25,28]. A large fraction of lncRNAs are predominantly localized in the nucleus, where they play critical roles in transcriptional and epigenetic regulation [24,28,29,30]. Recent studies have pointed to the involvement of this highly understudied class of RNAs in all aspects of tumor development and metastatic progression that transpires in essentially all human tumors [31,32,33,34,35,36]. However, nearly all mechanistic studies of lncRNA function have been performed using animal models or cultured cells employing knockdown and forced overexpression studies, which may not reflect their function in the context of a complex tumor milieu. Importantly, these studies are inherently incapable of addressing donor-to-donor variability inherent in human cancers.

We previously characterized and described the role of the murine lncRNA BORG (BMP2/OP1 Responsive Gene [37]) in breast cancer. Indeed, we found BORG expression to be aberrantly upregulated in human and murine TNBC cell lines and tumors, particularly those exhibiting aggressive metastatic and chemoresistant behaviors [38]. In the mouse, BORG (annotated as GM45924) is a Pol II-transcribed, nuclear-localized, spliced, and polyadenylated intergenic RNA [39]. Mechanistically, we showed that BORG binds to the E3 SUMO ligase TRIM28/KAP1 and that the resulting complex induces latent breast cancer cells to resume proliferative programs coupled to metastatic outgrowth and recurrence [38]. Moreover, BORG:TRIM28 complexes also govern the self-renewal and expansion of breast cancer stem cells, doing so by inducing the expression of Nanog, Aldh1a3, and Itga6/α6 integrins [40,41]. Finally, we observed BORG to interact physically with RPA1, an event coupled to the acquisition of resistance to doxorubicin via activation of the prosurvival NF-kB axis [42]. Despite these intriguing findings, a human gene corresponding to murine *BORG* has not been annotated or identified, and thus, the existence of a human ortholog for the mouse *BORG* gene was uncertain. In addressing this knowledge gap, the overall objectives of the current study were to (i) determine whether a human ortholog for BORG existed, and if successful, (ii) characterize the gene and define its genomic architecture, and (iii) define the function of human BORG in developing and progressing human breast cancers, particularly TNBCs, at the molecular level and in the context of large numbers of human tumor samples that account for donor-to-donor and tumor subtype variability.

## 2. Results

### 2.1. Identification and Architecture of Human BORG Genomic Locus

We identified the human BORG lncRNA through conservation of a small domain located close to the 3’ end of mouse BORG, which we have previously characterized [38,39,40,42]. Using this mouse-specific sequence in BORG, we now show additional conservation patterns between vertebrate species, syntenic preservation of its locus, and genomic direction of transcription and its 3′ processing region (Figure 1A and Appendix A). In humans, the *BORG* genic region is located on chromosome 8 (hg38 chr8: 103095156-103119540) and is transcribed in the antisense genomic direction. The human *BORG* gene is 24.3 kb long, and as observed in other lncRNAs, it is poorly spliced. Human *BORG* also contains a conserved 3′ end that lies in close vicinity to a canonical polyadenylation signal (Figure 1A and Appendix A). The transcriptional start site of human *BORG* overlaps a genomic region with H3K4Me1 and H3K27AC marks, consistent with the presence of an active/poised enhancer. Interestingly, this enhancer also gives rise to another lncRNA (LINC01181) through a bidirectional promoter, with the two transcription start sites less than 2000 nucleotides apart (Figure 1A). Finally, we confirmed that human BORG RNA lacks protein-coding capacity in a manner analogous to its murine ortholog (GM45924; Figure 1A, phyloCSF tracks). 

### 2.2. Human BORG RNA Is Strongly Upregulated in a Subset of Human TNBC Tumors

To determine whether human BORG RNA plays a role in the development and progression of breast cancers, we compared its expression pattern in normal versus cancerous breast tissue from human donors (Figure 1B and Appendix A). Our previous studies using human cell line models indicated that while the level of BORG expression was higher in all breast cancer subtypes, including ER-α, PR, and HER2 positive cell lines, the extent of upregulation was highest in cell lines representing the TNBC subtype [38]. Further analysis using primary breast cancers indicated that BORG expression in breast tumors was significantly elevated compared to breast organoids derived from non-cancerous primary breast cells (Figure 1B), especially in a subset of TNBC tumors. To confirm these results in a larger cohort of human patients, we took advantage of the breast cancer dataset of the Cancer Genome Atlas (TCGA BRCA) (Appendix A). Comparison of the BORG expression levels between distinct breast cancer histological and immunohistochemical subtypes (Appendix A) indicated that BORG expression was indeed dramatically elevated in a subset of human basal TNBC tumors (Figure 1A,C). Interestingly, despite the overall increased expression of BORG in basal TNBC tumors, we observed that within each histological group, including those expressing ER-α, PR, and HER2 (Appendix A), there is a range of BORG expression levels (Figure 1C). 

As a first step toward understanding the impact of BORG expression in human breast tumors, we compared tumors whose expression of BORG fell within the top and bottom quartiles of all breast cancers (BORG^High^ and BORG^Low^, respectively). Tumors from all four PAM50 subtypes (luminal A, luminal B, HER2^+^, and basal/TNBCs) were represented in both groups (Appendix A). We identified 2180 and 798 protein-coding genes that showed increased and decreased expression, respectively, in BORG^High^ tumors compared to their BORG^Low^ counterparts (>1.5-fold change and FDR < 0.05, Appendix A). Pathway analysis using the Hallmark gene lists of the Molecular Signatures Database (mSigDB) pointed to enrichment of pathways related to cellular proliferation and TGFβ signaling in BORG^High^ tumors (Figure 2A and Appendix A). Analysis of the expression pattern of genes known to be involved in aggressive cancer phenotypes indicated that most of these genes were indeed upregulated in BORG^High^ tumors (Figure 2B and Appendix A), including several genes associated with the basal phenotypes, EMT programs, pluripotency, and tumor invasion and metastasis [18,20,43,44,45] (Appendix A). Similarly, genes coding for multiple cytokines and surface protein markers that are known to be associated with the basal phenotype, pluripotency, and cancer aggression [14,15,20], such as CD44, KIT, EGFR, and PROM1, were also significantly upregulated in the BORG^High^ tumors (Appendix A). Consistent with these findings, our previous investigations analyzing mouse and human breast cancer cell lines both in vitro and in vivo clearly indicated that elevated expression of BORG is strongly associated with metastatic activity [38,40]. Collectively, these findings point to the intriguing possibility that increased levels of BORG may be associated with key carcinogenic pathways in human breast cancers of all subtypes. 

### 2.3. Increased Expression of BORG Is Associated with the Induction of Basal Transcriptomic Signatures

To compare the observed differences in BORG^High^ versus BORG^Low^ tumors to the transcriptomic patterns observed in cancers, we performed an enrichment analysis using the C2 gene lists from the mSigDB. Intriguingly, the top positively enriched gene list contained genes upregulated in basal breast cancers, while the most negatively enriched gene list comprised genes downregulated in basal breast cancers (Figure 2C). Multiple additional breast cancer-related gene lists were also included amongst the top 15 positively- and negatively enriched gene lists. These results strongly point to the association of higher BORG expression levels with gene expression programs found in basal tumors, including enrichment of proliferation-related pathways and a reduction in oxidative phosphorylation (Figure 3A, see also Appendix A). Interestingly, even after removing the basal tumors from the comparison groups, the pattern of changes in gene expression between the BORG^High^ and BORG^Low^ tumor groups remained largely unchanged (Appendix A and Figure 3A). Likewise, genes found in breast cancer progenitor cells were enriched in BORG^High^ tumors across all PAM50 tumor subtypes (Figure 3B–D), confirming our previous findings obtained in (i) mouse and human TNBC cell lines that BORG plays a significant role in governing the self-renewal and expansion of breast cancer stem cells [40], and (ii) metastatic human patient-derived xenografts that aberrant BORG expression associates with disease severity and the development of CNS metastases [38].

Taken together, these findings point to the presence of a shared transcriptomic signature associated with higher expression levels of BORG in all breast tumor subtypes, leading to increased proliferation, altered TGF-β signaling, and induction of progenitor phenotypes. However, it should also be noted that higher BORG expression was also associated with differential enrichment of many critical genes and pathways, such as those resulting in cancer aggression, upon inclusion of the basal/TNBC tumors in comparison groups (Figure 2B and Appendix A), but not when the non-basal breast cancer subtypes were independently analyzed (Appendix A). Thus, in addition to the effects that BORG elicits in all breast cancer subtypes, aberrant BORG expression may also drive additional aspects of the tumor gene expression program, specifically in the basal/TNBC breast cancer subtype.

### 2.4. Induction of Invasive Signatures in BORG^High^ TNBC Tumors

We next focused on defining the transcriptomic patterns specifically associated with elevated BORG expression within the basal/TNBC subset of breast cancers. Amongst the top 25 genes differentially expressed between TNBC tumors with the highest and lowest levels of BORG were TCF20, ARID1B, and UVSSA, all of which are involved in neoplastic processes such as pluripotency and DNA repair (Appendix A). We also identified several differentially expressed genes known to induce tumor invasiveness in breast cancers, the majority of which were upregulated in BORG^High^ TNBC tumors (Appendix A). The levels of several mRNAs that code for surface proteins with roles in cancer were similarly upregulated, including ALK, PTPRM, PTPRF, PTPRK, and SVEP1 (Appendix A). Interestingly, pathway analysis using the curated Hallmark gene lists indicated a strong induction of the EMT pathways, known to be involved in breast cancer invasiveness and metastasis; they also identified pathways involved in cellular proliferation and the TGFβ signaling (Figure 4A). As with pathway analyses in other breast cancer groups, oxidative phosphorylation was the top downregulated pathway (Figure 4A).

To obtain information on BORG-associated changes in a wider range of pathways and gene lists, we again used the C2 gene list database of mSigDB. Interestingly, multi-cancer invasive signature was the top enriched gene list in BORG^High^ tumors (Appendix A), and four additional breast cancer-related pathways were among the top 15 enriched gene lists (Appendix A). Amongst the subset of C2 pathways related to cancer, enriched pathways pointed to the association of high BORG expression levels with high-grade cancer phenotypes (Figure 4B,C). We showed that targets of a number of key carcinogenic transcription factors were enriched amongst genes upregulated in BORG^High^ tumors (Figure 4D), including those associated with metastatic and invasive phenotypes (FOXD1, EVI1/MECOM, MEF2C), EMT (EVI1/MECOM, OCT1/POU2F1, MEF2A), brain metastases (MEF2C), poor prognosis (FOXD1), and multi-drug resistance (NKX-2.5) [46,47,48,49,50,51,52,53] (Figure 4D–F). Interestingly, our previous work in murine models of breast cancer development and metastatic progression demonstrated robust acquisition of pro-metastatic and drug-resistance phenotypes following BORG overexpression [38,40,42]. Collectively, these findings indicate that higher BORG expression levels in basal/TNBC tumors are associated with pathways involved in induction of highly invasive phenotypes, including those operant in regulating EMT, metastasis, and drug-resistance. 

### 2.5. Aberrant BORG Expression Leads to Induction of Genes Associated with Aggressive Basal Phenotypes

To determine whether increased BORG expression plays a causative role in the induction of basal/TNBC phenotypes, we leveraged RNA-seq analyses of the D2.HAN breast cancer model that consists of indolent D2.OR cells that express little-to-no BORG and their aggressive D2.A1 counterparts that endogenously express robust quantities of BORG [38,40,41,42]. Importantly, heterologous expression of murine BORG in D2.OR cells confers aggressive and metastatic phenotypes reminiscent of those observed in their D2.A1 counterparts (Appendix A) [38,42]. RNA-seq analyses of parental and BORG-expressing D2.OR cells identified 387 protein-coding genes that were consistently differentially expressed as a result of aberrant BORG expression. Mapping these differentially expressed genes to pathways and gene lists involved in breast cancer identified multiple positively and negatively enriched pathways. The top enriched pathways activated by BORG included those we observed to be enriched in BORG^High^ human breast cancer tumors, including the basal breast cancer and breast cancer progenitor gene lists (Appendix A). Taken together, these findings suggest that increased BORG expression plays a causative role in eliciting the formation of aggressive basal breast cancers.

To further investigate this intriguing possibility, we compared the genes that were differentially expressed following heterologous BORG expression in D2.OR cells to those upregulated in BORG^High^ human breast tumors. Interestingly, over 270 of the 387 genes that were differentially expressed in BORG-overexpressing D2.OR cells were also differentially expressed in the same direction in BORG^High^ human TNBC tumors as compared to their BORG^Low^ counterparts (Figure 5A). These results suggest that at least part of the gene expression difference between BORG^High^ and BORG^Low^ TNBC tumors is likely the direct result of higher BORG expression levels in these tumors. Interestingly, these BORG-associated genes corresponded to those upregulated in basal breast cancers (e.g., Smid breast cancer basal up gene list) and key cancer-related processes and pathways, such as oncogenesis, metastasis, and stress response (Figure 5A,B and Appendix A), many of which were also enriched in BORG^High^ breast tumors (Appendix A, Figure 2C and Figure 3A). These intriguing results indicate that the BORG-induced gene expression pattern plays a key role in induction of the aggressive, pro-metastatic basal phenotype, which is strongly associated with the TNBC tumors. 

Finally, we also studied the expression pattern of BORG-dependent genes in the CMI/MBC cohort of the TCGA, which includes paired primary and metastatic breast cancers from multiple donors (n = 6 pairs). Genes differentially expressed in paired metastatic versus primary tumors included many of the BORG-dependent genes. Further, these genes mapped to gene lists that overlapped with those upregulated in BORG^High^ TNBCs, and in D2.OR cells engineered to express BORG (e.g., Smid breast cancer basal up gene list, Figure 5C). These findings further point to the relevance of this group of genes in cancer aggression and the metastatic process. To directly assess the impact of BORG on the expression of this group of genes, we compared BORG expression levels versus the aggregate expression levels of genes associated with the basal breast cancer phenotype (Smid breast cancer basal up gene list) in the TCGA cohort of breast tumors. Importantly, in basal TNBC tumors, higher levels of BORG showed direct correlation with the aggregate expression score of the Smid breast cancer basal up genes (Pearson correlation coefficient of 0.35 for the entire basal TNBC cohort (n = 118) and 0.49 for tumors in the top three quartiles in terms of BORG expression) (Figure 5D and Appendix A). Interestingly, many of these genes are BORG-dependent genes (Figure 5A and Appendix A), and as such, these findings indicate that BORG expression is likely to be a major driving force for induction of the basal gene expression pattern in human breast cancer. 

## 3. Discussion

In this study, we provide the first identification of the human ortholog of the murine BORG lncRNA. In addition to characterizing its locus, we further define the impact of changes in the expression of human BORG observed in large cohorts of human breast tumors. These analyses show that BORG is aberrantly elevated in primary human breast tumors compared to normal human mammary tissue. Importantly, there is significant variability in BORG expression levels across all breast cancer subtypes. As we had described for the murine BORG RNA, higher expression levels of human BORG in tumors are also associated with the induction of invasive, metastatic, and “stemness” gene signatures commonly observed in metastatic tumors [38,40,41,42]. Accordingly, a significant fraction of genes differentially expressed in BORG-overexpressing murine TNBCs show concordant expression changes when BORG^High^ human tumors were compared to their BORG^Low^ counterparts. It is noteworthy that BORG, like many other studied lncRNAs, is poorly conserved in its primary sequence [23,24,25], having only two relatively short (300–400 nucleotides long) stretches of evolutionarily conserved sequences in its genic region. However, as observed for many other poorly conserved lncRNAs [26,27], the function of BORG in the human and mouse TNBC tumors is strongly conserved. 

Our study of the protein interactome of the mouse BORG indicates that it binds TRIM28/KAP1, an E3 Sumo ligase known to be involved in regulation of pluripotency-related pathways [38,54,55]. Interestingly, in both mice and humans, increased expression of BORG is associated with increased breast cancer stem cell characteristics in the tumors. Indeed, aberrant expression of BORG in mouse and human breast cancers was associated with the acquisition of breast cancer stem cell-like properties and upregulation of known breast cancer stem cell markers, such as NANOG, ALDH1, and ITGA6/*α*6Integrin/CD49f. Thus, these findings indicate that induction of stemness properties in breast cancers is a conserved function of BORG, thereby explaining the observed association of BORG with metastatic properties in tumors. The role of BORG in the initiation and progression of TNBCs has not yet been directly studied. However, based on existing data, it is possible to speculate that elevated BORG expression leads to improved cellular survival and stem-like phenotypes [38,40,42]. This, in turn, provides a survival advantage to cells with elevated BORG expression levels while they circumnavigate functional bottlenecks as they traverse the metastatic cascade. Further, due to its pro-survival effect at the cellular level, BORG strongly contributes to resistance to chemotherapeutic agents [42]. 

Our analysis comparing high versus low BORG-expressing tumors indicates that across all breast cancer subtypes, higher BORG levels were associated with enhanced stem cell/progenitor phenotypes and increased expression of genes that are typically associated with basal tumors. This finding points to the presence of a core set of pathways that are induced in all breast cancer subtypes in association with elevations in BORG expression. On the other hand, several key oncogenic pathways associated with aggressive breast cancer behavior, such as EMT and invasive signatures, were either uniquely or much more prominently induced in BORG^High^ basal/TNBC tumors compared to other breast cancer subtypes analyzed in this study. Taken together, these findings point to a subtype-specific impact for increased BORG expression, thereby creating a nuanced picture of BORG function in breast cancer with potential clinical and therapeutic relevance. 

As mentioned above, there is strong similarity between the transcriptomic signature of enforced BORG expression in murine TNBC models versus the gene expression patterns observed when high and low BORG-expressing TNBC tumors are compared. This, in turn, implies a causative role for BORG in inducing at least a fraction of the gene expression patterns observed in high BORG tumors. Over 70% of genes differentially expressed in mouse TNBC cells engineered to overexpress BORG were concordantly differentially expressed in BORG^High^ human TNBC tumors. Further, BORG-expressing mouse TNBCs induced an overlapping set of cellular pathways when compared to BORG^High^ human tumors. We showed that a significant fraction of the transcriptomic signature associated with aberrant BORG expression is mediated through induction of expression and/or activation of a handful of transcription factors, particularly OCT1, MEF2A, and MEF2C that function in regulating EMT and pluripotency. It is plausible that BORG, in association with its complement of RNA-binding proteins, may directly affect chromatin modification or transcriptional activity at MEF2C and OCT1 loci, thus setting a pro-metastatic, pro-invasiveness program into motion. Taken together, the above results point to the existence of a human ortholog for BORG RNA and a key pro-metastatic function for human BORG. By establishing the overall importance of BORG in activating pro-metastatic cellular pathways and defining the inter-donor and subtype variability of BORG activity, our findings establish BORG as a unique target for therapeutic development and as a potentially potent biomarker to guide the choice of therapeutic strategy. 

## 4. Methods

### 4.1. BORG Overexpression Studies in Mouse Cell Lines

Murine D2.HAN (D2.OR, RRID:CVCL_0I88; and D2.A1, RRID:CVCL_0I90) cells used in this study were obtained from Dr. Fred Miller (Wayne State University, Detroit, MI, USA). The cell lines were propagated in DMEM media (Sigma-Aldrich, St. Louis, MO, USA) that was supplemented with 10% FBS and 1% Pen/Strep. D2.OR cells were authenticated using short tandem repeat analysis (ATCC, Manassas, VA, USA), and were regularly tested for mycoplasma infection using the MycoAlert Mycoplasma Detection Kit (Lonza, catalog #LT07-218, Basel, Switzerland). D2.OR cells used in the described studies were passaged less than 20 times before data acquisition. 

D2.OR cells were stably transfected with a plasmid carrying the full-length mouse BORG cDNA as a transgene, and the expression of the transgene was validated by real-time RT-PCR for BORG as described [38]. Total cellular RNA was obtained from parental (empty vector) and BORG-expressing D2.OR cells for bulk RNA-seq analysis. For each experimental condition, three technical replicates were included. 

### 4.2. Bulk RNA-Seq Analyses

RNA-seq studies were performed on parental (empty vector) and BORG-expressing D2.OR derivatives that were propagated in 3D-culture for 7 days, followed by harvesting total cellular RNA from the resulting organoids. Sample preparation was performed as described [42]. RNA sequencing reactions were performed on a HiSeq 2500 Illumina platform instrument yielding an average of ~66 million paired end, 100 nucleotide long reads for each sample. The raw sequencing data has been deposited in the NCBI SRA (accession number below).

Quality of the resulting sequenced reads was assessed using FastQC version 0.11.9 (Babraham Bioinformatics, Cambridge, UK), followed by pre-processing using Trim Galore version 0.6.6 (Babraham Bioinformatics), which is a wrapper based on Cutadapt and FastQC, to remove any leftover adaptor-derived sequences, as well as sequences with Phred scores less than 30. Any reads shorter than 40 nucleotides after the trimming were not used in alignment. Subsequently, the trimmed reads were aligned to the mouse genome (*Mus musculus*, GRCm38/mm10) using Kallisto v0.48.0 [56], followed by normalization using Sleuth v0.30.1 [57]. Pairwise differential expression tests were performed using generalized linear models as implemented in edgeR version 3.42.4 (QL) [58], and false discovery rate (FDR) values were calculated for each differential expression value. Protein-coding genes that were expressed at a minimum abundance of 5 transcripts per million (TPM) were used for pathway analysis with fold-change values as the ranking parameter that was controlled against FDR at 0.05. The Gene Set Enrichment Analysis (GSEA, version 4.0.2) package was used to identify the enriched pathways and promoter elements using mSigDB version 7.0 and KEGG version 100.0 databases. Pathways showing an FDR q-value <= 0.25 were considered significantly enriched, per the GSEA package guidelines. The number of genes contributing to the enrichment score was calculated using the leading-edge output of GSEA (tag multiplied by size).

The human bulk RNA-seq data used in this study were downloaded from SRA (bioproject accession PRJNA227137) [59] or The Cancer Genome Atlas [4] (TCGA-BRCA and CMI-MBC). The quality control and pre-processing steps were performed as above. The pre-processed reads were aligned to the human genome (hg38/GRCh38) using Gencode release 28 as the reference annotations with STAR version 2.7.2b [60], followed by gene-level quantitation using Htseq-count version 0.11.3 [61]. In parallel, the pre-processed reads were pseudo-aligned using Kallisto (v0.43.1 [56]), with 100 rounds of bootstrapping to the Gencode release 28 of the human transcriptome. The resulting values were normalized using Sleuth, with the two pipelines yielding concordant results. The genomic alignments were used for defining the transcriptional architecture of the *BORG* locus. 

### 4.3. Identification of the Human BORG Locus

For defining the human *BORG* locus, de novo transcriptome assembly packages RefShannon version 0.0.1 [62] and Cufflinks version 2.2.1 [63] were used. Expression patterns of nearby genes were evaluated to ensure that the reads assigned to the *BORG* locus were not resulting from run-through transcriptional activity from nearby genes. Samples that showed run-through patterns were eliminated from analysis. The direction of transcriptional activity at the human *BORG* locus was determined using strand-specific RNA-seq studies. Representative TNBC and non-TNBC tumors used in read histogram representations were identified as the samples in which the expression of BORG was closest to the mean BORG expression value within each group. 

### 4.4. Filtering and Preparation of the Study Cohort

The TCGA BRCA dataset was filtered to eliminate all samples that had a high percentage of normal, non-cancerous cells. Only tumor samples that were unequivocally assigned to a single breast cancer subtype (e.g., basal TNBC, luminal A, luminal B, or HER2^+^) were used in the study. Tumors that did not have a histological type (PAM50 subtype) assigned were excluded from these analyses. Similarly, due to their small number, non-basal TNBC tumors were also excluded from this study. PAM50 subtype assignments were further verified using gene expression analysis to monitor ERBB2, ESR1, and basal and luminal cytokeratin expression patterns. Tumors labeled as metastatic or secondary, which formed a very small group of tumors in this study, were eliminated from the analysis. The triple-negative status of TNBC cells was verified using both RNA expression patterns and protein expression data included in the TCGA data repository. 

## 5. Conclusions

An accumulating body of work in the scientific literature has established lncRNAs as playing critical roles in the pathogenesis and progression of human breast cancers. The recent development and implementation of antisense drugs opens the door to therapeutically targeting oncogenic lncRNAs. BORG, with its dual impact on the induction of metastatic activity and resistance to chemotherapy, is a highly promising target for such therapies. Importantly, BORG is expressed in very few healthy tissues, and outside the central nervous system, its expression is only detectable in the kidneys at very low levels. This, in turn, suggests that targeting BORG using antisense therapeutics may prove to be highly effective and safe due to a lack of off-target activity. Taken together, our studies not only reveal a critical role for BORG in driving the development of aggressive, drug-resistant basal TNBC phenotypes but also highlight its potential as a promising therapeutic target for the treatment of the most aggressive breast tumors. 

## Figures and Tables

**Figure 1 cancers-16-03241-f001:**
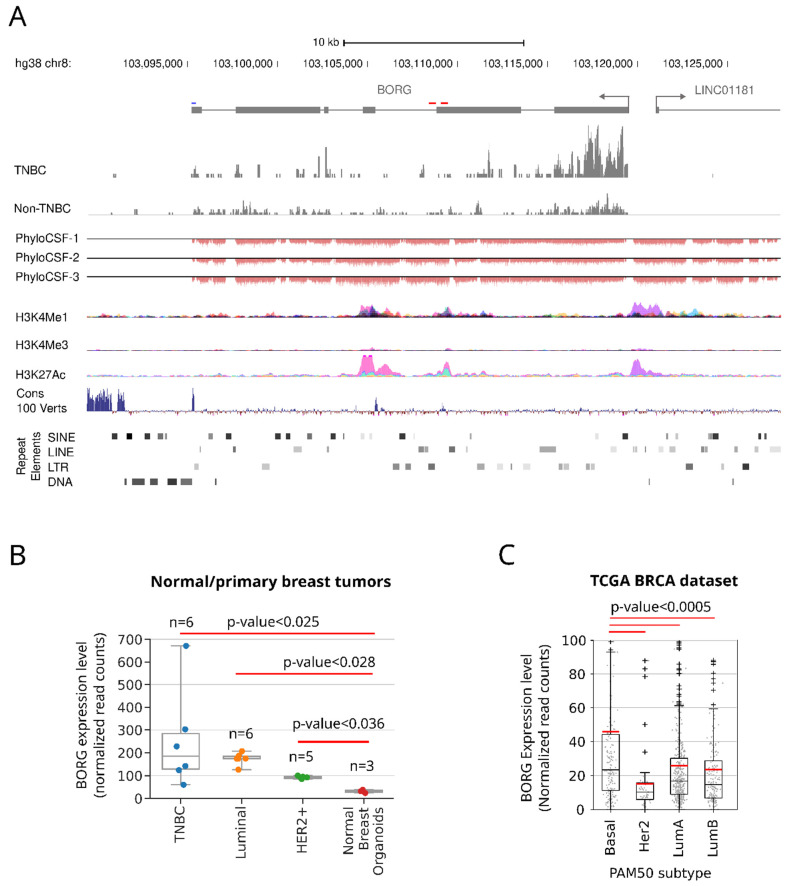
The genomic locus, architecture, and expression pattern of human BORG RNA. (**A**) The genomic locus of the human *BORG* RNA and its exon–intron architecture are shown. The position of the transcription start site of a divergent lncRNA (LINC01181) that originates from a shared promoter with *BORG* is marked. The short red lines above the gene model indicate the regions conserved between the mouse and human *BORG*. The blue line marks a conserved sequence close to the 3′ end of the human BORG that likely contributes to its 3′ end processing. This sequence is in close vicinity of a canonical polyadenylation signal sequence located 30 nucleotides upstream of the 3′ end of BORG transcript, which is shown in Appendix A. Read histograms for representative tumors from TNBC and non-TNBC groups are shown. PhyloCSF tracks, shown in the three reading frames corresponding to the direction of transcription of BORG, indicate the lack of protein-coding capacity in human BORG RNA. Tracks showing markers of enhancer elements from the UCSC genome browser indicate the presence of an enhancer element close to the 5′ end of *BORG*. Conservation tracks point to the conserved regulatory regions close to the 3′ end of BORG and the area conserved between mouse and human. The position of repeat elements is shown at the bottom. (**B**) BORG expression is strongly induced in breast cancers, especially in TNBC tumors, as compared to normal mammary organoids. Dots within the box plots mark the expression level of individual samples. (**C**) BORG is highly expressed in a subset of human breast cancers of the basal TNBC subtype. Red and black horizontal lines indicate the mean and median, respectively. Gray dots within the box plots mark the expression level of individual tumors.

**Figure 2 cancers-16-03241-f002:**
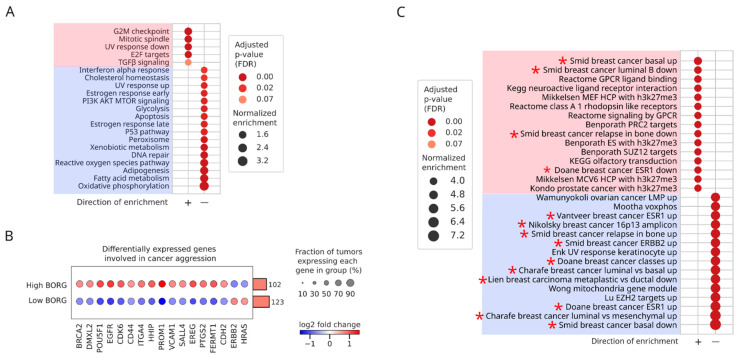
Elevated expression of BORG is associated with increased cancer aggression markers and the induction of basal phenotypes. (**A**) Comparison of differential pathway enrichment patterns between breast cancer samples falling in the upper (BORG^High^) versus lower (BORG^Low^) quartiles with respect to BORG expression levels amongst breast cancers of all histological subtypes. Molecular Signatures Database (mSigDB) Hallmark gene lists are used. Significantly enriched pathways in the positive (pink rectangle) versus negative (blue rectangle) directions are shown. Proliferation-related terms and TGFβ signaling, known to be involved in cancer aggressiveness, are significantly induced. (**B**) Several genes known to associate with aggressive breast cancer phenotypes are upregulated in BORG^High^ compared to BORG low tumors. The rotated bar plot to the right of the dot plot indicates the number of tumors in each group. (**C**) Pathway analysis as described in (**A**), using C2 gene lists of mSigDB, indicates a strong induction of a basal phenotype among BORG^High^ tumors. Breast cancer-related pathways are marked by an asterisk.

**Figure 3 cancers-16-03241-f003:**
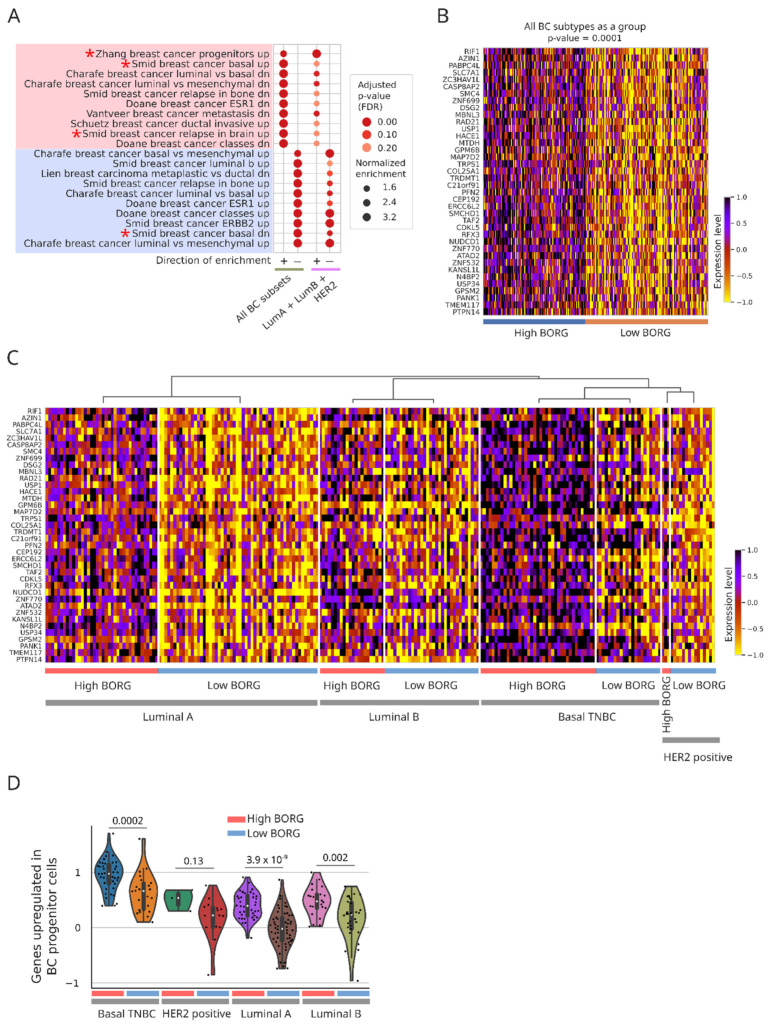
Many gene expression patterns associated with higher BORG expression levels are shared between all TNBC subtypes. (**A**) Pathway analysis on genes differentially expressed in BORG^High^ versus BORG^Low^ tumors of non-basal breast cancer subtypes as a group. These results point to the association of basal transcriptomic signatures with high BORG expression, even in non-basal breast cancer subtypes. Interestingly, genes found in breast cancer progenitor cells were strongly enriched in non-basal tumors with high BORG expression levels. Key pathways and gene lists are marked with asterisks. (**B**,**C**) Heatmap of genes constituting those associated with breast cancer progenitors indicates their specific upregulation in BORG^High^ tumors in all breast cancer subtypes, including basal, shown as a group (**B**) and separately for each subtype (**C**). (**D**) Aggregate scores for genes upregulated in breast cancer progenitor cells indicate their overall increase in BORG^High^ tumors in all breast cancer subtypes. Each dot represents a tumor. p-values are shown at the top. For HER2+ tumors, the number of tumors was too small to derive a reliable p-value.

**Figure 4 cancers-16-03241-f004:**
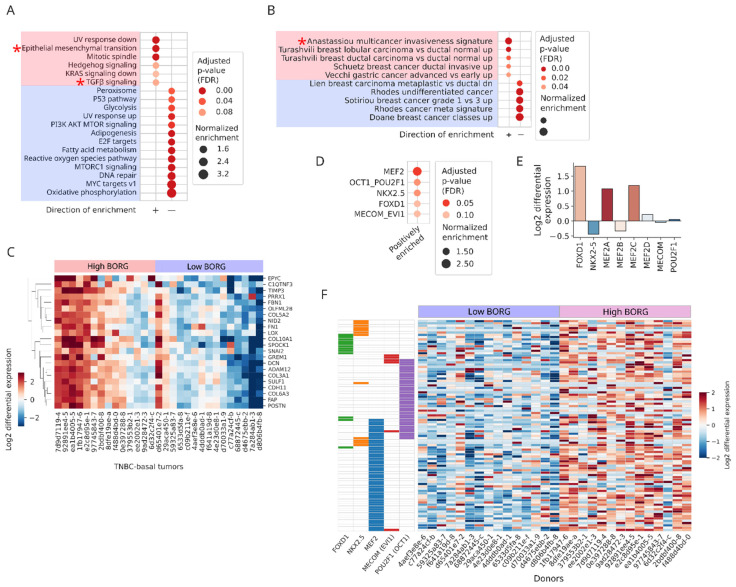
Higher expression of BORG in basal TNBC tumors is associated with induction of invasiveness and pro-metastatic gene expression programs. (**A**) Pathway analysis comparing BORG^High^ versus BORG^Low^ basal TNBC tumors points to the induction of the EMT and TGFβ pathways. (**B**) Using the cancer-related gene lists from the mSigDB C2 database of genes points to a set of genes comprising a multi-cancer invasiveness signature as the top enriched gene list in basal TNBCs (marked by an asterisk). (**C**) Multi-cancer invasiveness signature genes (shown on the left of the heatmap) are strongly upregulated in BORG^High^ TNBC tumors compared to their BORG^Low^ counterparts. Donor IDs are shown at the bottom. (**D**) Transcription factor binding motif analysis indicates that genes induced in BORG^High^ basal TNBC tumors are enriched for targets of multiple pro-neoplastic, pro-metastatic transcription factors. (**E**) The pro-metastatic MEF2 and FOXD1 transcription factors are transcriptionally upregulated in BORG^High^ TNBCs. (**F**) Genes regulated by the five transcription factors in Figure 3D are upregulated in BORG^High^ TNBC tumors, indicative of a global induction of a pro-metastatic gene expression program.

**Figure 5 cancers-16-03241-f005:**
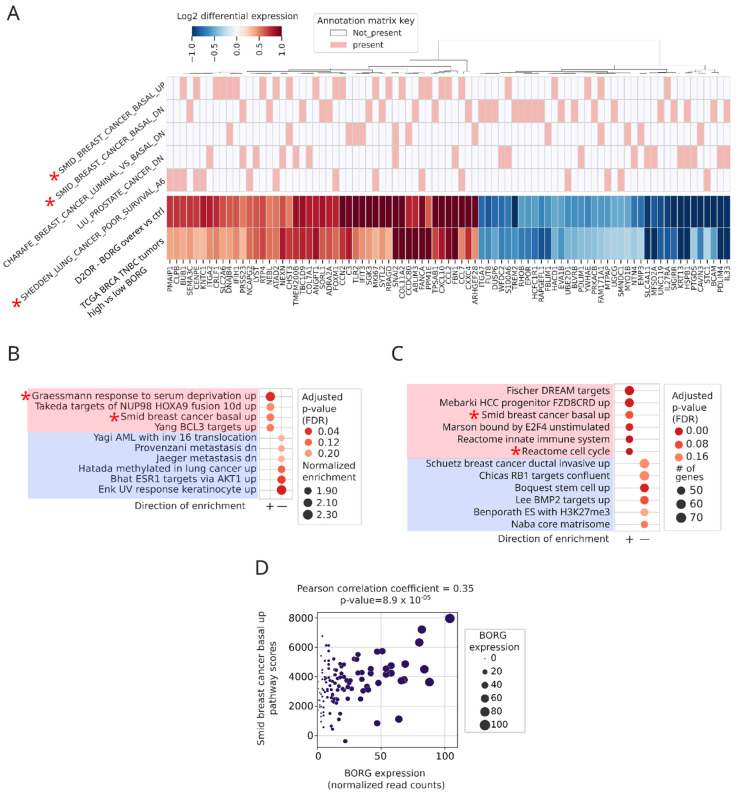
Increased expression of BORG has a causative role in induction of an aggressive, basal-like transcriptomic signature in breast tumors. (**A**) D2.OR cells, which overexpress BORG from a transgene, and BORG^High^ TNBC tumors share differentially expressed genes that include those upregulated in basal breast cancer. Heatmap at the bottom shows shared differentially expressed genes that map to the top five most enriched pathways. Rows contain the differential expression pattern of genes when BORG-overexpressing cells are compared to empty vector-transfected cells (D2.OR-BORG Overex versus ctrl) and those that are differentially expressed in BORG^High^ versus BORG^Low^ TNBCs (TCGA BRCA TNBC tumors high versus low BORG). The annotation matrix at the top maps each gene to a specific pathway. Identity of the pathways/gene lists is shown to the left. Asterisks mark gene lists associated with poor outcomes and aggressive basal breast cancer signatures. (**B**) Pathway analysis using the C2 pathway list on genes that show concordant differential expression between D2.OR BORG-overexpressing cells and BORG^High^ basal TNBC tumors. The results show the most positively enriched pathways among these shared genes include Smid breast cancer basal up gene list, consistent with BORG-mediated induction of expression of basal-specific gene expression patterns. (**C**) Comparison of paired primary and metastatic breast cancers from the same donor (n = 6 pairs) identifies multiple upregulated gene lists and pathways in metastatic tumors, including genes upregulated in basal tumors (Smid breast cancer basal up). (**D**) The extent of upregulation of genes induced in basal breast cancer (Smid breast cancer basal up genes) is positively correlated with the level of expression of BORG in TNBC tumors. Pearson’s correlation coefficient for the entire group of basal TNBC tumors (n = 118) is shown. In tumors with BORG expression in the top three quartiles, the correlation coefficient was 0.49 (p-value 7.32 × 10^−6^), indicating a stronger correlation between BORG level and the basal phenotype at higher BORG expression levels. The aggregate expression score of the Smid breast cancer basal up genes was calculated as the average expression of the set of genes subtracted with the average expression of a reference set of genes. The reference set is randomly sampled from the gene expression table for each binned expression value.

## Data Availability

The raw sequencing files for the mouse data presented in this study are available in NCBI Sequence Read Archive (SRA) at https://www.omicsdi.org/dataset/geo/GSE99234 (accessed on 17 September 2024).

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
