# Peer review of "Induction of Invasive Basal Phenotype in Triple-Negative Breast Cancers by Long Noncoding RNA BORG"

_cancers, 2024, doi:10.3390/cancers16183241_

Round 1
Reviewer 1 Report
Comments and Suggestions for Authors
General comment
- The article "Induction of invasive basal phenotype in triple-negative breast cancers by long noncoding RNA BORG" is highly interesting. Understanding the understudied long noncoding RNA (lncRNA) in triple-negative breast cancer is important as it may provide insights into the genetically related progression of TNBC and may be used as a potential target for therapy.
- Even though need some additional assays, the authors addressed the methodology section, results, and discussion in a clear and concise manner.
Minor comment
- In order to verify the invasion and migration assays for the transfection of suitable cells, it may be necessary to determine whether overexpression of BORG is associated with invasion and metastasis. This information is lacking in the present study. This outcome might raise readers' interest and the quality of the article. Although studies have demonstrated that overexpression of BORG leads to the upregulation of several types of mRNA relevant to invasion and metastasis, these genes have been associated to other activities that may not be directly related to invasion and metastasis. Therefore, further data regarding the function of BORG in TNBC invasion and metastasis may be obtained through direct measurement.
- The authors discussed on the function of BROG in relation to human TNBC and its subtypes. However, recommend using human TNBC cells in their results and methodology section, or provide an explanation for using solely murine cells or may modify the topic related to murine model.
Author Response
Point by point response to reviewers’ comments
We thank the reviewers for their constructive comments which have made the manuscript stronger and improved the flow of the text. Below is a detailed response to reviewer comments, with our responses highlighted in gray.
Reviewer 1
General comment
- The article "Induction of invasive basal phenotype in triple-negative breast cancers by long noncoding RNA BORG" is highly interesting. Understanding the understudied long noncoding RNA (lncRNA) in triple-negative breast cancer is important as it may provide insights into the genetically related progression of TNBC and may be used as a potential target for therapy.
- Even though need some additional assays, the authors addressed the methodology section, results, and discussion in a clear and concise manner.
- We thank the reviewer for the positive evaluation of our work and for the constructive comments and suggestions, which we have addressed below.
Minor comment
- In order to verify the invasion and migration assays for the transfection of suitable cells, it may be necessary to determine whether overexpression of BORG is associated with invasion and metastasis. This information is lacking in the present study. This outcome might raise readers' interest and the quality of the article. Although studies have demonstrated that overexpression of BORG leads to the upregulation of several types of mRNA relevant to invasion and metastasis, these genes have been associated to other activities that may not be directly related to invasion and metastasis. Therefore, further data regarding the function of BORG in TNBC invasion and metastasis may be obtained through direct measurement.
- We thank the reviewer for bringing up this important point. We agree that providing information about these important aspects of BORG function will help further clarify the function of BORG. To this end, we have added the following text to the manuscript (lines 174-177 of the revised manuscript, highlighted in yellow):
“Consistent with these findings, our previous investigations analyzing mouse and human breast cancer cell lines both in vitro and in vivo clearly indicated that elevated expression of BORG is strongly associated with metastatic activity38,40. ”
- The authors discussed on the function of BORG in relation to human TNBC and its subtypes. However, recommend using human TNBC cells in their results and methodology section, or provide an explanation for using solely murine cells or may modify the topic related to murine model.
- We have addressed this issue by adding the following text (lines 206-210 of the revised manuscript, highlighted in yellow):
“confirming our previous findings obtained in (i) mouse and human TNBC cell lines that BORG plays a significant role in governing the self-renewal and expansion of breast cancer stem cells40, and (ii) metastatic human patient-derived xenografts that aberrant BORG expression associates with disease severity and the development of CNS metastases38.”
Green highlights mark sentences edited for clarity/improved flow.
Reviewer 2 Report
Comments and Suggestions for Authors
This manuscript entitled "Induction of invasive basal phenotype in triple-negative breast cancers by long noncoding RNA BORG" describes the role of a long noncoding RNA (lncRNA) called BORG in triple-negative breast cancer (TNBC). While lncRNAs are known to play key roles in the onset and progression of cancer, detailed studies on the specific function of BORG in TNBC are scarce. Therefore, this study provides new insights into this field.
M & M : The experimental methods used in the paper are valid and well-described. The authors have analyzed the expression pattern of BORG in various breast cancer datasets and investigated its role in TNBC using the D2.HAN breast cancer model. This experimental design is highly useful for a deep understanding of the function of BORG.
Results : The results section is well-written, with data supported by figures and tables. The authors demonstrate that the expression of BORG plays a significant role in inducing aggressive phenotypes in TNBC. They also show that increased expression of BORG activates pathways related to cancer invasiveness, metastasis, and drug resistance. These findings are important contributions to our understanding of TNBC.
Minor
While this study makes a significant contribution to our understanding of BORG’s role, there are still several issues to be addressed. The specific mechanisms by which BORG influences the onset and progression of TNBC are not fully understood. Also, the impact of increased BORG expression on the treatment of TNBC needs to be investigated. Additional experiments and research are needed to address these issues. These aspects require detailed explanation in the discussion section.
In figure legend, some sentences are somewhat long and complex, which could make them difficult to understand. These sentences could be revised for clarity and conciseness.
Comments on the Quality of English Language
The manuscript is generally well-written with no major grammatical errors.
Author Response
Reviewer 2
This manuscript entitled "Induction of invasive basal phenotype in triple-negative breast cancers by long noncoding RNA BORG" describes the role of a long noncoding RNA (lncRNA) called BORG in triple-negative breast cancer (TNBC). While lncRNAs are known to play key roles in the onset and progression of cancer, detailed studies on the specific function of BORG in TNBC are scarce. Therefore, this study provides new insights into this field.
M & M : The experimental methods used in the paper are valid and well-described. The authors have analyzed the expression pattern of BORG in various breast cancer datasets and investigated its role in TNBC using the D2.HAN breast cancer model. This experimental design is highly useful for a deep understanding of the function of BORG.
Results : The results section is well-written, with data supported by figures and tables. The authors demonstrate that the expression of BORG plays a significant role in inducing aggressive phenotypes in TNBC. They also show that increased expression of BORG activates pathways related to cancer invasiveness, metastasis, and drug resistance. These findings are important contributions to our understanding of TNBC.
- We thank the reviewer for the positive comments on our work and the constructive points raised, which we have addressed as described below each comment.
Minor
While this study makes a significant contribution to our understanding of BORG’s role, there are still several issues to be addressed. The specific mechanisms by which BORG influences the onset and progression of TNBC are not fully understood. Also, the impact of increased BORG expression on the treatment of TNBC needs to be investigated. Additional experiments and research are needed to address these issues. These aspects require detailed explanation in the discussion section.
- To address these points, as suggested by the reviewer, we have added the following text to the Discussion section (lines 373-378 of the revised manuscript, highlighted in yellow):
“The role of BORG in initiation and progression of TNBCs has not yet been directly studied. However, based on existing data, it is possible to speculate that elevated BORG expression leads to improved cellular survival and stem-like phenotypes38,40,42. This, in turn, provides a survival advantage to cells with elevated BORG expression levels while they circumnavigate functional bottlenecks as they traverse the metastatic cascade. Further, due to its pro-survival effect at cellular level, BORG strongly contributes to resistance to chemotherapeutic agents42. ”
In figure legend, some sentences are somewhat long and complex, which could make them difficult to understand. These sentences could be revised for clarity and conciseness.
- We thank the reviewer for alerting us to this issue. We have gone through the legends for both main and supplementary figures and revised the figure legends to improve their readability. We also adopted a standardized nomenclature (BORGhigh and BORGlow) for referring to tumors that fell in the upper and lower quartiles in terms of BORG expression level throughout the manuscript. This helped shorten the sentences and make them easier to read. These modified sentences are highlighted in green to differentiate them from the rest of the changes made in response to reviewer and editor comments in the manuscript, which are highlighted in yellow.
Reviewer 3 Report
Comments and Suggestions for Authors
This manuscript the authors present an interesting study on the ncRNa BORG
while the in silico analysis of TCGA datasets and others show promising data, the amount of relevant followup on the mechanistic actions of BORG remain completely untested.
the authors show that basal like and TNBC tumors tend to have the highest expression of BORg, however the authors did not look directly at ERalpha status, which is the number 1 molecular diagnostic for all of clinical breast cancer .diadnosis.
the authors also mentioned normal breast organoids but not tumor organoidd
the authors did not even compare 231 vs Mcf-7 cells
i would suggest further experimentation that interrogates mechanism of
action before resubmitting this paper
Comments on the Quality of English Language
N/a
Author Response
Reviewer 3
This manuscript the authors present an interesting study on the ncRNa BORG
while the in silico analysis of TCGA datasets and others show promising data, the amount of relevant followup on the mechanistic actions of BORG remain completely untested.
- We have revised the manuscript to include additional discussion of the work previously done by us using mouse models and mouse and human cell lines. We hope that this help clarify that the goal of this manuscript has been to confirm our previous findings, which were obtained in study models, in a large cohort of primary human tumors. Importantly, in this manuscript we aimed to address the crucial but frequently-overlooked issue of variability observed between human tumors due to the stage of the disease and inter-donor variability. These additions have been highlighted in yellow in the Results and Discussion sections.
The authors show that basal like and TNBC tumors tend to have the highest expression of BORG, however the authors did not look directly at ERalpha status, which is the number 1 molecular diagnostic for all of clinical breast cancer diagnosis.
- To address this concern, we have further clarified in the text that the expression level of ER-alpha has been studied in supplementary figure 4, along with that of the progesterone receptor and HER2-positive status of all the tumors studied in the TCGA BRCA cohort at both protein and RNA levels. Specifically, we have modified the text (lines 156-158 in the revised manuscript) which now reads:
“Interestingly, despite the overall increased expression of BORG in basal TNBC tumors, we observed that within each histological group, including those expressing ER-a, PR, and HER2 (Fig. S4C, D), there is a range of BORG expression levels (Fig. 1C). ”
Please also see Fig. S4.
The authors also mentioned normal breast organoids but not tumor organoid. The authors did not even compare 231 vs Mcf-7 cells.
- We have addressed this point in the revised manuscript by referring the readers to our manuscript (Gooding et al., 2017) where the expression of BORG is studied in normal breast tissue along with cancerous tissues obtained from multiple study models. The added text is located at line 149 of the revised manuscript and is highlighted in yellow.
I would suggest further experimentation that interrogates mechanism of action before resubmitting this paper
- We agree with the reviewer that including such information will help strengthen the manuscript. As discussed above and in response to similar suggestions from the reviewers 1 and 2, we have added new text that clarifies the outcome of our mechanistic studies performed in mouse and human TNBC models to help the reader correlate them with data obtained here from a large, diverse cohort of human primary tumors.
Round 2
Reviewer 3 Report
Comments and Suggestions for Authors
I thank the authors for their comments.
however it seems to me that if the group already published data using preclinical animal
models, why are they only presenting in silico analysis? I was hoping the authors would have added some additional mechanisms in at a minimum cultures cell lines between era+ and era- cells
Comments on the Quality of English LanguageN/a
Author Response
Reviewer 3:I thank the authors for their comments. however it seems to me that if the group already published data using preclinical animal models, why are they only presenting in silico analysis? I was hoping the authors would have added some additional mechanisms in at a minimum cultures cell lines between era+ and era- cells".
Our response:
In our previous point-by-point response document accompanying the revised manuscript, and within the revised manuscript itself, we indicated that we have already performed and published the very experiments and findings that the reviewer has asked us to perform. For instance, our study published in 2017 (see Gooding et al., 2017, DOI:10.1038/s41598-017-12716) clearly demonstrated the differences in BORG expression and function between ER-positive and ER-negative breast cancers (Fig. 1A-C). It is important to note that requiring us to republish these findings constitutes data duplication, which is a problematic form of scientific misconduct. We are certain that the reviewer, similar to us, strongly believes is abiding by the scientific rules of conduct. We reiterate that the objective of the present manuscript is to complement our extensive, previously published mechanistic studies by (i) fully characterizing the human BORG locus and (ii) analyzing how its aberrant activation is linked to the development and progression of breast tumors across a large cohort of primary human breast cancers using computational methods. We respectfully remind the reviewer that we submitted our manuscript to the Cancer Informatics and Big Data section of the journal, for publication in the Application of Bioinformatics in Cancers collection. The recently published manuscripts in this collection, similar to ours, do not include any wet lab components. We fail to see why our manuscript should be held to different standards.
As the requested experimental data has already been published by us, we were able to address the request of the reviewer in a satisfactory manner by adding a sentence (lines 149-152) which clarifies that these studies have been performed and published by us, citing the relevant reference.